# Synthesis and Characterization of a Novel Resveratrol Xylobioside Obtained Using a Mutagenic Variant of a GH10 Endoxylanase

**DOI:** 10.3390/antiox12010085

**Published:** 2022-12-30

**Authors:** Ana Pozo-Rodríguez, Juan A. Méndez-Líter, Rocío García-Villalba, David Beltrán, Eva Calviño, Andrés G. Santana, Laura I. de Eugenio, Francisco Javier Cañada, Alicia Prieto, Jorge Barriuso, Francisco A. Tomás-Barberán, María Jesús Martínez

**Affiliations:** 1Department of Microbial and Plant Biotechnology, Centro de Investigaciones Biológicas Margarita Salas, Spanish National Research Council (CIB, CSIC), C/Ramiro de Maeztu 9, 28040 Madrid, Spain; 2Department of Quality, Safety and Bioactivity of Plant Foods, Centro de Edafología y Biología Aplicada del Segura, Spanish National Research Council (CEBAS, CSIC), Espinardo, 30100 Murcia, Spain; 3Department of Structural and Chemical Biology, Centro de Investigaciones Biológicas Margarita Salas, Spanish National Research Council (CIB, CSIC), C/Ramiro de Maeztu 9, 28040 Madrid, Spain; 4CIBER de Enfermedades Respiratorias (CIBERES), Avda. Monforte de Lemos 3-5, 28029 Madrid, Spain; 5Department of Chemistry of Natural and Synthetic Bioactive Products, Instituto de Productos Naturales y Agrobiología, Spanish National Research Council (IPNA, CSIC), Avda. Astrofísico Francisco Sánchez 3, 38206 San Cristóbal de La Laguna, Spain

**Keywords:** antioxidants, colonic fermentation, fungal enzyme, glycoconjugates, glycoside hydrolases, glycosynthase, polyphenols, solubility, xylobiose

## Abstract

Resveratrol is a natural polyphenol with antioxidant activity and numerous health benefits. However, in vivo application of this compound is still a challenge due to its poor aqueous solubility and rapid metabolism, which leads to an extremely low bioavailability in the target tissues. In this work, rXynSOS-E236G glycosynthase, designed from a GH10 endoxylanase of the fungus *Talaromyces amestolkiae,* was used to glycosylate resveratrol by using xylobiosyl-fluoride as a sugar donor. The major product from this reaction was identified by NMR as 3-*O*-ꞵ-d-xylobiosyl resveratrol, together with other glycosides produced in a lower amount as 4′-*O*-ꞵ-d-xylobiosyl resveratrol and 3-*O*-ꞵ-d-xylotetraosyl resveratrol. The application of response surface methodology made it possible to optimize the reaction, producing 35% of 3-*O*-ꞵ-d-xylobiosyl resveratrol. Since other minor glycosides are obtained in addition to this compound, the transformation of the phenolic substrate amounted to 70%. Xylobiosylation decreased the antioxidant capacity of resveratrol by 2.21-fold, but, in return, produced a staggering 4,866-fold improvement in solubility, facilitating the delivery of large amounts of the molecule and its transit to the colon. A preliminary study has also shown that the colonic microbiota is capable of releasing resveratrol from 3-*O*-ꞵ-d-xylobiosyl resveratrol. These results support the potential of mutagenic variants of glycosyl hydrolases to synthesize highly soluble resveratrol glycosides, which could, in turn, improve the bioavailability and bioactive properties of this polyphenol.

## 1. Introduction

Resveratrol (*trans*-3,5,4′-trihydroxystilbene) is a natural polyphenol that belongs to the stilbenoids group and is present in many plants and fruits, such as peanuts, mulberries, blueberries, raspberries and, especially, in grapes and red wine [1,2]. It acts as a phytoalexin, protecting the plants from pathogens, mechanical injuries and UV radiation [3]. Numerous in vitro studies have demonstrated the antioxidant properties of resveratrol as well as other health benefits including antitumor and anti-inflammatory activities, cardiovascular and neuronal protection and even antiaging effects, which makes resveratrol a promising compound for the pharmaceutical, nutraceutical and cosmetic industries [3,4]. However, in vivo application of this compound is still a challenge due to its poor aqueous solubility (0.03–0.05 g·L^−1^) and extensive metabolism [1,3,5,6]. As previously reported, resveratrol absorption after an oral dose is very high, around 75% [1,7]. Unfortunately, the rapid metabolic conversion that takes place in the small intestine and liver results in a very low resveratrol concentration in blood, thereby reducing its bioavailability in the target tissues to less than 1% [7,8,9]. In order to circumvent these limitations, a novel approach aiming to seek or synthesize resveratrol derivatives with enhanced solubility and bioavailability has been established [3,8].

One potential approach to obtain resveratrol derivatives is based on glycosylation [2,3]. It is well documented that the glycosylation of phenols (e.g., epigallocatechin gallate, phloretin, pterostilbene and hydroxytyrosol) can improve their solubility, bioavailability, stability, safety and even their bioactive properties [10,11,12,13,14,15]. Two glycosylated derivatives of resveratrol, known as piceid (3-*O*-ꞵ-d-glucosyl resveratrol) and resveratroloside (4′-*O*-ꞵ-d-glucosyl resveratrol), have been identified in the Japanese knotweed *Polygonum cuspidatum* [2,3]; however, extractions from natural sources are affected by harvest variability and the yields obtained are very low [16]. Chemical glycosylation could be an alternative [17,18], but it generates toxic by-products and requires many protection and de-protection steps to avoid the lack of regio- and stereoselectivity [19,20]. Therefore, the enzymatic glycosylation of resveratrol arises as a promising strategy for the efficient and sustainable production of these glycoconjugates [21]. Nevertheless, the number of resveratrol glycosides that have been enzymatically synthesized up to now is still small, with few examples of glucosides [22,23,24], maltosides [23] and other hexose-derived glycosides [25,26], and no resveratrol pentosylglycosides (e.g., xylosides or xylobiosides) described.

Glycoside hydrolases (GHs) are a large and heterogeneous group of enzymes [27] that catalyze the hydrolysis of glycosidic linkages, being powerful tools for the valorization of lignocellulosic residues [28]. In certain conditions, some of these enzymes also display the capacity of transferring glycosyl residues to acceptor compounds in a reaction called transglycosylation [29]. Nonetheless, the transglycosylation yields obtained with wild-type GHs are generally low and a strict control of the reaction is required to prevent the enzymes from hydrolyzing the glycosylated products [29,30]. To overcome these drawbacks, their hydrolytic capacity should be eliminated or drastically reduced, enhancing their synthetic activity [29,30]. Directed mutagenesis of the catalytic nucleophile residue by an inert amino acid is the most successful approach to obtain glycosynthases [30], which will require the addition of a fluorine-activated donor with inverted anomeric configuration to synthesize glycoconjugates and oligosaccharides of different lengths [20,31].

The ascomycetous fungus *Talaromyces amestolkiae* stands out for its ability to secrete high levels of GHs [28]. These enzymes were reported to be promising tools to break down plant biomass, but some of them also catalyze transglycosylation reactions and were applied to synthetize the glycosides of bioactive phenolic compounds [13,32,33]. This is the case of the versatile endoxylanase XynSOS, recently expressed in the yeast *Pichia pastoris* (rXynSOS) and subsequently characterized [34]. With the aim of improving the yields of transglycosylation products, the glycosynthase variant rXynSOS-E236G was designed by directed mutagenesis and successfully expressed in *P. pastoris*. This enzyme variant, which lacked hydrolytic activity, showed a broad acceptor spectrum that includes phenolic antioxidants such as gallic acid, phloretin and rosmarinic acid, and the potential to synthesize xylooligosaccharides of 4, 6, 8 and 10 xylose units [34].

In this work, we describe the glycosylation of resveratrol catalyzed by the fungal glycosynthase rXynSOS-E236G and the characterization of its major reaction product, 3-*O*-ꞵ-d-xylobiosyl resveratrol. The synthesis of this glycoconjugate was optimized, and its antioxidant activity and solubility were analyzed. In addition, as this xylobioside is not expected to be metabolized in the proximal gastrointestinal tract [35], we performed colonic fermentations with human fecal samples to test if it could be hydrolyzed in the colon, enabling the release of free resveratrol which, as described by other authors [35,36,37,38,39,40], will subsequently be converted into other bioactive metabolites.

## 2. Materials and Methods

### 2.1. Production and Purification of rXynSOS-E236G Glycosynthase

rXynSOS-E236G glycosynthase from *T. amestolkiae* was heterologously produced in *P. pastoris* GS115 and purified as previously reported [34]. Seven-day-old cultures grown in 1 L flasks with 200 mL of YEPS medium at 28 °C and 250 rpm with daily addition of 6.5 mL·L^−1^ methanol were used for enzyme purification. Cultures were centrifuged for 20 min at 10,000× *g* and 4 °C (Sorvall LYNX 6000 centrifuge, Thermo Scientific, Waltham, MA, USA) and the supernatant was sequentially vacuum filtered through 0.8, 0.45 and 0.22 μm nitrocellulose membrane discs (Millipore). Then, it was concentrated and dialyzed against 10 mM phosphate buffer pH 6 using an ultrafiltration cell (Amicon, Millipore) with a 10 kDa cutoff polysulfone membrane (Millipore). rXynSOS-E236G glycosynthase was purified in a single chromatographic step with an Äkta FPLC system (Cytiva). A 5 mL QFF HiTrap anion exchanger cartridge (Cytiva) was equilibrated with 10 mM phosphate buffer pH 6.0 and ran with a flow of 2 mL·min^−1^. The enzyme was not retained in the column, which was washed with 2.5 column volumes of 1 M NaCl in 10 mM phosphate buffer pH 6.0 to remove retained proteins. The system was finally re-equilibrated to the initial conditions. After purification, the enzyme was concentrated by ultrafiltration in 10 kDa cutoff Amicon Ultra-15 centrifugal devices (Millipore). Protein concentration was determined using a Nanodrop spectrophotometer (Thermo Fisher Scientific) and by Bradford assays (Bio-Rad Laboratories).

### 2.2. Resveratrol Glycosylation Assays

The standard resveratrol glycosylation reaction catalyzed by rXynSOS-E236G was carried out at pH 5, the optimum for rXynSOS [34] in 50 mM sodium acetate buffer with 20 mM xylobiose-fluoride (X_2_F) as the xylobiose donor, 10 mM resveratrol (dissolved in 20% acetonitrile) as the acceptor and 4 mg·mL^−1^ of enzyme. The reaction (100 μL) was conducted for 16 h at 25 °C and 1000 rpm. Resveratrol was acquired from Shangai Seebio Biotechnology (Shanghai, China). The X_2_F donor contains the fluorine atom in α configuration at carbon 1 of xylobiose and was synthesized as previously reported [41]. Several solvents in different *v*/*v* percentages (10–50%) were also tested before choosing 20% (*v*/*v*) acetonitrile. The solvent list included acetone, ethanol, methanol, dimethyl sulfoxide (DMSO) and dimethylformamide (DMF).

### 2.3. Analysis of Glycosylated Products by TLC and ESI-MS

To reveal the synthesis of resveratrol glycosides, the reactions were examined by thin-layer chromatography (TLC) using silica gel G/UV254 polyester plates (Macherey-Nagel) and a mobile phase of ethyl acetate:methanol:water in 10:2:1 (*v*/*v*). The plates were first visualized under 254 nm UV light and then by immersion on a solution of 5% sulfuric acid in methanol followed by heating for 12 min at 100 °C.

To identify the products observed in TLC, the corresponding reactions were analyzed by electrospray ionization-mass spectrometry (ESI–MS) in an HCT ultra ion trap (Bruker Daltonics) using methanol as the ionizing phase in the positive and negative reflector modes. The data were processed with the Masshunter Data Acquisition B.05.01 and Masshunter Qualitative Analysis B.07.00 softwares (Agilent Technologies). The expected glycosides and xylooligosaccharides were detected as sodium adducts in the positive mode and resveratrol was observed as its hydrogen adduct in the negative mode.

### 2.4. Analysis and Quantification of Resveratrol Glycosides by HPLC

The production and conversion (%) of the resveratrol glycosides synthesized by rXynSOS-E236G were calculated by HPLC in an Agilent 1200 series LC instrument equipped with a reverse phase ZORBAX Eclipse plus C18 column (Agilent). The column was equilibrated in a mix of acetonitrile (ACN) and H_2_O (27:73, *v*/*v*) with 0.1% acetic acid at a flow of 2 mL·min^−1^. The reaction products were separated isocratically in 8 min. After elution, the column was washed for 5 min with 95:5 (*v*/*v*) ACN:H_2_O, and re-equilibrated to the initial conditions for 3 min. Potential glycosides were detected by monitoring the absorbance at 270 nm and quantified by comparing peak areas with the calibration curve of non-glycosylated resveratrol.

### 2.5. Purification and NMR Analysis of Resveratrol Glycosides

The main fractions containing resveratrol glycosides, tagged as glycoside **1** and glycoside **2**, were further isolated and studied by NMR. Both fractions were separated by HPLC, using the settings described in Section 2.4, and later lyophilized. For NMR analyses, samples of the reference compounds, resveratrol, X_2_F and xylobiose, and the purified glycosides were prepared by dissolving them in deuterated water to concentrations around 1.5–3 mM. NMR spectra were acquired at 298 K, using a Bruker AVIII 600 MHz spectrometer (Bruker BioSpin). 1D ^1^H, 2D ^1^H-^13^C HSQC and HMBC, ^1^H-^1^H ROESY (300 ms mixing time) and TOCSY (20 and 70 ms mixing time) and DOSY (Diffusion Ordered SpectroscopY) experiments were performed to assign all NMR signals and to characterize their molecular size (DOSY). For the 1D ^1^H, 2D ^1^H-^13^C HSQC and HMBC, DOSY, ROESY and TOCSY experiments, the standard zg, zgpr, hsqcedetgp, hmbcgpndqf, ledbpgp2s, roesyphpr and dipsi2phpr sequences included in TOPSPIN software (Bruker Daltonics) were employed. For DOSY, pseudo 2D experiments with 16 different field gradient intensities increasing linearly from 2% to 98% were acquired using a 140 ms diffusion delay (big delta, d20) and 1400 μs of gradient pulses time (little delta, p30). Chemical shifts were referenced to the residual water signal set at 4.77 ppm at 298 K [42].

### 2.6. Optimization of Resveratrol Glycosylation by Response Surface Methodology

The reaction conditions for the synthesis of the major resveratrol xylobioside (3-*O*-ꞵ-d-xylobiosyl resveratrol) were optimized by applying a response surface methodology approach. A Box–Behnken design matrix was created with the Design-Expert^®^ software version 10.0.1.0 (Stat-Ease Inc. Minneapolis, MN, USA), using the following parameters and ranges: X_2_F concentration (5–30 mM), resveratrol concentration (2–10 mM), enzyme dosage (0.5–4 mg·mL^−1^) and reaction time (30 min–5 h). The rest of the parameters, including pH, temperature and acetonitrile percentage, were fixed to pH 5, 25 °C and 20%, respectively. After conducting the required reactions, they were analyzed by HPLC as indicated in Section 2.4, to calculate the production and conversion (%) of 3-*O*-ꞵ-d-xylobiosyl resveratrol. Then, the software generated polynomial quadratic equations from the experimental data, which show the effect of the independent variables on the response of the process, as well as the highest production and conversion (%) expected. The models developed for resveratrol glycosylation were validated by performing the reactions predicted to reach the maximum production and conversion (%).

### 2.7. Solubility Assay

Saturated aqueous solutions of resveratrol and 3-*O*-ꞵ-d-xylobiosyl resveratrol were prepared in MilliQ water and maintained for 30 min at room temperature (RT) and 1000 rpm. Then, the solutions were centrifuged, filtered and analyzed by HPLC, using the protocol described in Section 2.4, to determine the concentration of each molecule. When needed, the saturated solutions were diluted in MilliQ water to fit the calibration curve of non-glycosylated resveratrol.

### 2.8. Antioxidant Activity Assay

The 2,2-azino-bis(3-ethylbenzo-thiazoline-6-sulfonic acid) diammonium salt (ABTS^•+^) method was used to determine the antioxidant activity of resveratrol and its xylobioside by adding 6-hydroxy-2,5,7,8-tetramethylchroman-2-carboxylic acid (Trolox, Merck), as a reference antioxidant compound. The ABTS^•+^ was generated by incubating 3.5 mM ABTS (Roche) with 1.22 mM potassium persulfate (Sigma-Aldrich) for 16 h at RT in the dark and diluting it with ethanol to obtain an absorbance of ~0.7 at 734 nm. This solvent was chosen instead of water due to the insolubility of resveratrol and its low interference with the assay. Solutions of increasing concentrations of Trolox (7.8–1000 μM), and resveratrol and 3-*O*-ꞵ-d-xylobiosyl resveratrol (3.9–3000 μM), were prepared in ethanol. The assay was then performed in triplicate in 96-well plates by mixing 10 μL of each antioxidant with 115 μL of ABTS^•+^ working solution. The reduction in ABTS^•+^ absorbance was monitored with a SpectraMax^®^ M2 microplate reader (Molecular Devices, San Jose, CA, USA) for 15 min by measuring at 734 nm every minute. The following formula was applied to represent the reduction in ABTS^•+^ absorbance:$$\mathrm{ABTS}{\;}^{•+}\;\mathrm{reduction}\;\left(\%\right)=\left(\frac{\mathrm{Abs}{\;}_{\mathrm{control}}-{\;\mathrm{Abs}\;}_{\mathrm{antioxidant}}}{{\mathrm{Abs}\;}_{\mathrm{control}}}\right)\times \;100$$

SigmaPlot (Stat-Ease, Minneapolis, MN, USA) was used to plot the curves of ABTS^•+^ reduction (%) and to calculate the EC50 (μM) of each compound, which refers to the concentration of antioxidant needed to reduce the ABTS^•+^ initial absorbance by 50%. Trolox equivalent antioxidant capacity (TEAC) was also determined to show the concentration of the antioxidant (μM) that reduces ABTS^•+^ absorbance to the same extent as 1 μM of Trolox. The TEAC values were calculated using the EC50 of resveratrol or its xylobioside and the EC50 of Trolox. A *t*-test was also conducted with SigmaPlot to estimate the significance (*n* = 3, *p* < 0.001) of the EC50 and TEAC values of the xylobioside compared to those of non-glycosylated resveratrol.

### 2.9. Colonic Fermentations

Colonic fermentations were carried out by using human fecal samples provided by two healthy volunteers who followed a normal diet and declared not having ingested antibiotics for at least 3 months before sample collection. Preparation of fecal suspension and subsequent culturing experiments were conducted under anoxic conditions in an anaerobic chamber (Don Whitley Scientific Limited, Shipley, UK) with an atmosphere consisting of N_2_/H_2_/CO_2_ (80:10:10) at 37 °C. Fecal samples (10 g) were diluted 1/10 (*w*/*v*) in Nutrient Broth medium (NB, Merck), supplemented with 0.06% L-cysteine hydrochloride and homogenized with a stomacher in filter bags. An amount of 50 µL of filtered fecal suspensions were inoculated into 5 mL of fermentation medium (anaerobe basal broth, ABB, Oxoid) to grow colonic bacterial strains, and 40 µL of 5 mM 3-*O*-ꞵ-d-xylobiosyl resveratrol prepared in water were added. The tubes were incubated in anaerobic chambers at 37 °C to mimic colon conditions. Samples were collected at 0, 2, 4, 6, 8, 10, 12 and 24 h. Two controls (culture medium without 3-*O*-ꞵ-d-xylobiosyl resveratrol and culture medium with 3-*O*-ꞵ-d-xylobiosyl resveratrol but without fecal inocula) were also prepared for each fermentation time point. The incubations were carried out in the level 2 safety laboratory at CEBAS-CSIC, equipped to handle human biological samples.

After incubation, 5 mL of the culture medium were extracted with 5 mL of ethyl acetate and the samples were centrifuged at 3500× *g* and 4 °C for 10 min. The supernatants (the upper layer) were evaporated in a speed vacuum concentrator, re-dissolved in 250 µL of methanol and filtered through a 0.22 µm filter before injection for HPLC analysis. This fraction mainly contained resveratrol aglycone. The remaining aqueous fraction after removing the ethyl acetate layer was passed through a Sep-Pack C18 column previously conditioned first with 10 mL of methanol and then with 10 mL water. The aqueous fraction was then filtered through the activated cartridge to retain resveratrol xylobioside and other polar metabolites. Polar derivatives of resveratrol were then eluted with 2 mL of methanol that was evaporated, and samples were reconstituted in 250 µL of methanol and filtered through a 0.22 µm filter before injection in the HPLC equipment. The production of resveratrol and the remaining resveratrol xylobioside were then analyzed by HPLC-DAD-MS, using a system (1200 Series, Agilent Technologies) equipped with a photodiode-array and single quadrupole mass spectrometer detectors in series (6120 Quadrupole, Agilent Technologies). Chromatographic separations were performed in a Poroshell 120 EC-C18 column (3 × 100 mm, 2.7 μm) (Agilent Technologies), using, as mobile phases, 0.5% formic acid in water (A) and acetonitrile (B) with a flow rate of 0.5 mL·min^−1^. A gradient elution was applied: 0−7 min with 5−18% B, 7−17 min with 18−28% B, 17−22 min with 28−50% B and 22−27 min with 50−90% B, and this was maintained for 1 min and then taken back to the initial conditions. Samples of 5 μL were injected onto the column and the analyses were performed at RT. Quantification was performed by UV detection at 280 nm, by comparison with authentic standards of resveratrol (Sigma-Aldrich) and the synthesized resveratrol xylobioside to follow the kinetics of production during fecal fermentations.

## 3. Results and Discussion

### 3.1. Glycosylation of Resveratrol by rXynSOS-E236G Glycosynthase

The ability of *T. amestolkiae* glycosynthase rXynSOS-E236G to produce resveratrol glycosides was tested using X_2_F as a sugar donor. Due to its extremely low aqueous solubility, the amount of resveratrol that could be added to the reactions without precipitating was very low. To make the glycosylation reactions feasible, several cosolvents were assessed, ruling out the use of ethanol and methanol as they are potential glycosylation acceptors of the glycosynthase. We first observed that the glycosylation yield was higher with acetonitrile than with acetone, DMSO or DMF (data not shown), selecting this solvent and assaying concentrations between 10% and 30% (*v*/*v*). Further reactions were finally performed with 20% acetonitrile (*v*/*v*), as it allowed the use of higher resveratrol concentrations without compromising the glycosynthase activity (data not shown).

Then, the standard reaction mixture was prepared with 10 mM resveratrol and 20 mM X_2_F in 20% acetonitrile, with 4 mg·mL^−1^ of rXynSOS-E236G glycosynthase. After overnight incubation, potential resveratrol glycosides were first detected by TLC (Figure 1A). An intense spot and a diffuse band, which could correspond to a mixture of glycoconjugates, were observed as glycosylation products in the reactions with the biocatalyst. The ESI-MS analysis confirmed these results, since the mass of the compound detected in the major peak matched with that of the sodium adduct of the resveratrol xylobioside (Figure 1B). Longer glycosides, containing 4 and 6 xylose units attached to the resveratrol, were also detected as sodium adducts. This suggests that rXynSOS-E236G produces several resveratrol glycoconjugates, connecting xylobiose molecules to different -OH groups of resveratrol or forming longer carbohydrate chains in a single position. In addition, the identification of xylotetraose as one of the reaction products confirmed the previously reported potential of the enzyme to produce xylooligosaccharides [34].

### 3.2. Purification and NMR Analysis of Glycosylated Products

The reaction mixture was also analyzed by HPLC to determine the glycosylation profile of resveratrol and to purify the products when possible. The chromatogram showed that this glycosynthase generated a complex glycosylation profile with different product peaks (Figure 2). A major peak that eluted at 2.38 min was labeled as glycoside **1**, and the minor peak at 1.97 min as glycoside **2**. Finally, several early eluting small peaks could not be further separated and were assumed to be a mixture of minor resveratrol glycosides since they were detected at 270 nm.

The material eluted in the two main product peaks separated by HPLC (glycosides **1** and **2**) was analyzed by NMR for structural analysis. Regarding regioselectivity, there are three potential glycosylation sites in resveratrol (3, 5 and 4′), which can yield glycosides with different properties. In the case of glycoside **1**, the glycosynthase attached the xylobiosyl residue to the -OH group at position 3 of the resveratrol (Figure 3A), as derived from the observation of the correlations in the ROESY spectrum of the anomeric H1 of the xylobiose to the protons in positions 2 and 4 in the resveratrol (Appendix A), yielding the 3-*O*-ꞵ-d-xylobiosyl resveratrol.

On the other hand, for glycoside **2,** the results pointed to a mixture of two different glycosides. The most abundant of them was characterized as xylobiose linked to the 4′-OH group of resveratrol (Figure 3B), as deduced from the ROESY crosspeak between the anomeric H1 of the xylobiose and the protons on the carbons 3′ and 5′ of the resveratrol (Appendix A), rendering the 4′-*O*-ꞵ-d-xylobiosyl resveratrol. The minor component turned out to be a glycoside of a higher molecular mass, as derived from the DOSYU [43] spectrum (Appendix A), tentatively assigned to the 3-*O*-ꞵ-d-xylotetraosyl resveratrol (Figure 3C), which is feasible since rXynSOS-E236G is capable of synthesizing xylooligosacharides [34]. Within the glycoside **2** mixture, the 4′-*O*-ꞵ-d-xylobiosyl resveratrol was three times more abundant than the 3-*O*-ꞵ-d-xylotetraosyl resveratrol (Appendix A), as determined from the relative integrals of the peaks detected at 7.58 ppm (H2’ and H6’ protons of the 4′-*O*-ꞵ-d-xylobiosyl resveratrol) and 7.52 ppm (H2’ and H6’ protons of the 3-*O*-ꞵ-d-xylotetraosyl resveratrol). The three deduced glycoside structures are depicted in Figure 3 and their corresponding spectra and chemical shifts can be found in Appendix A, respectively.

To the best of our knowledge, this is the first time that these three resveratrol glycosides have been produced, since there are no reports available describing the enzymatic synthesis of resveratrol xylobiosides or other pentose-glycosylated derivatives. On the contrary, the production of α-glucosides and α-maltosides catalyzed by β-cyclodextrin glucanotransferases [22,23] and amylosucrases [24] has already been reported. Moreover, β-diglucosides and β-triglucosides have also been synthetized with UDP glucosyltransferases [25,26], as well as glycosides from other hexoses such as galactosides, rhamnosides and fucosides [25].

The study of resveratrol derivatives has attracted a lot of attention because they could enhance the solubility and bioavailability of the aglycon and even improve its bioactivity. For instance, the water solubility of piceid, one of the known natural resveratrol glucosides, widely overpasses that of the non-glycosylated molecule and has a better effect on murine non-alcoholic fatty liver disease [44,45]. In addition, it has been reported that the β-glucosylation and β-maltosylation of resveratrol improved its antiallergic and vasodilatory activities in rat peritoneal mast cells [46]. Therefore, further analysis of the properties of the three novel resveratrol glycosides synthesized by rXynSOS-E236G glycosynthase seems to be extremely interesting. To obtain an amount large enough to undertake these analyses, we tried to optimize the synthesis of the main reaction product, 3-*O*-ꞵ-d-xylobiosyl resveratrol.

### 3.3. Optimization of the Synthesis of 3-O-ꞵ-d-Xylobiosyl Resveratrol by Response Surface Methodology

The production of 3-*O*-ꞵ-d-xylobiosyl resveratrol was optimized by applying a Box–Behnken design response surface method for modelling parameters, such as donor and acceptor concentrations, enzyme dosage and reaction time, while the optimum pH and temperature were fixed (pH 5 and 25 °C, respectively). The production (mM) and conversion (%) for 3-*O*-ꞵ-d-xylobiosyl resveratrol were the responses selected for the optimization.

The matrix of experiments generated by the Design Expert software comprised 29 reactions and the results regarding the productions and conversions (%) obtained in each of them were used to generate a polynomial quadratic model whose behavior is described by the equations available in the Appendix A. The experimental setup was validated by an analysis of the variance test carried out by the software.

The model describes that the highest productions (~3.2 mM 3-*O*-ꞵ-d-xylobiosyl resveratrol) are obtained with the maximum resveratrol concentration of 10 mM, elevated or moderated X_2_F concentrations and different combinations of enzyme dosages and reaction times. This agrees with the results previously published for other glycosynthases [13,15]. Regarding the conversion (%), there is no clear condition that can be related to the best results (~32% conversion to 3-*O*-ꞵ-d-xylobiosyl resveratrol), pointing to an equilibrium between the resveratrol and X_2_F concentration. In contrast to the data previously reported [13,15], low acceptor and high donor concentrations drastically decreased the conversion (%) for the 3-*O*-ꞵ-d-xylobiosyl resveratrol, while the conversion (%) for the other potential resveratrol glycosides eluted in the early peaks of the chromatogram shown in Figure 2 was enhanced (data not shown). The high versatility of the enzyme might explain this finding because when all the -OH groups in position 3 of resveratrol are already glycosylated but X_2_F is still available in the reaction mixture, the glycosynthase can attach a second xylobiose molecule to a different -OH group of resveratrol or to the previously bound xylobiose, thereby forming new glycosides.

Finally, the multiparametric model was used to predict the maximum production and conversion (%) for the 3-*O*-ꞵ-d-xylobiosyl resveratrol within the selected limits of the parameters. Thus, the maximum production that can be reached (Figure 4A) is 3.48 mM in a 1.27 h reaction, using nearly 10 mM of resveratrol as the acceptor, 26.33 mM X_2_F as the donor and 3.76 g·L^−1^ of rXynSOS-E236G glycosynthase. The maximum conversion (%) of resveratrol to its major xylobioside (Figure 4B) is 35.28%, in a reaction of 0.64 h with 9.63 mM of resveratrol, 24.29 mM of X_2_F and 3.99 g·L^−1^ of rXynSOS-E236G. These reactions were carried out to validate the predictions, yielding a production and conversion (%) such as the predicted ones.

The use of this response surface methodology approach was successful, since the optimized reaction yielded around a 35% conversion of the resveratrol into 3-*O*-ꞵ-d-xylobiosyl resveratrol and a total conversion of the polyphenol close to 70%, summing up all the glycosylated products. As this is the first time that a resveratrol xylobioside has been synthesized, no data are available to compare the conversion (%) obtained. However, we can analyze these results considering the conversion (%) reached for the resveratrol glycosides produced from other carbohydrates. For instance, the conversions for the 4′-*O*-α-d-glucosyl resveratrol and 3-*O*-α-d-glucosyl resveratrol obtained in a 12 h reaction catalyzed by an amylosucrase were 38.7% and 6.8%, respectively [24]. Moreover, the α-glycosylation of resveratrol conducted in 2 h by a cyclodextrin glucanotransferase resulted in a total conversion of 35% and a mixture of glucosides and maltosides [23]. On the other hand, a 24 h transglycosylation with a cyclodextrin glucanotransferase produced a 50% total conversion and several α-glucosides as products [22]. Finally, a UDP glucosyltransferase synthesized four different resveratrol glucosides, glycosylating 90% of the substrate in 12 h [25]. Thus, the conversion (%) for the 3-O-ꞵ-D-xylobiosyl resveratrol as well as the total conversion (%) of resveratrol to the mixture of the glycosides obtained with the rXynSOS-E236G glycosynthase are among the best reported so far and were achieved in shorter reaction times, which could reduce the cost of the process.

### 3.4. Effect of Xylobiosylation on the Antioxidant Activity of Resveratrol

The antioxidant activity of resveratrol and the major xylobioside produced by the glycosynthase were analyzed by using the ABTS^•+^ reduction method, and Trolox as the reference molecule. The Trolox equivalent antioxidant capacity (TEAC) values for both compounds can be found in Table 1. Resveratrol has a TEAC value lower than 1, meaning that it shows a higher antioxidant activity than Trolox. This is not the case with 3-*O*-ꞵ-d-xylobiosyl resveratrol, which suffers a significant decrease in its antioxidant capacity (2.21-fold) compared to the aglycon. Nevertheless, it could still be considered a strong antioxidant molecule comparable to Trolox as it has a TEAC value of approximately 1. This loss of antioxidant activity due to the glycosylation of phenols is expectable, since their antioxidant power is related to the presence of free -OH groups, capable of donating hydrogens to scavenge reactive oxygen free radicals. Therefore, as some -OH groups are now involved in glycosidic bonds, the antioxidant capacity of the parent molecule is reduced [47,48]. In fact, the glycosides of other phenolic compounds, such as vanillyl alcohol [49], phloretin [11] and EGCG [10,15], also underwent this reduction with respect to their aglycons. Nevertheless, at least for the glucose-based glycoconjugates, the initial levels of antioxidant activity could be restored once the glycoside is hydrolyzed in vivo and the free phenolic precursor is released [11,50,51,52,53,54]. An assay devoted to evaluating whether the xylobiosides produced in this work are deglycosylated in the large intestine will be presented in Section 3.6.

Several studies have analyzed the antioxidant capacity of various resveratrol glycosides, to determine the relevance of the different -OH groups of the aglycon on this bioactivity. The ABTS^•+^ method was used to compare 3-*O*-α-d-glucosyl resveratrol and 4′-*O*-α-d-glucosyl resveratrol, detecting antioxidant activities 2.26 and 1.46 times lower, respectively, than the non-glycosylated molecule [22]. This could imply that the -OH located at position 3 of resveratrol is more important for its antioxidant activity than the 4′-OH. Similar results were reported for the same glycosides measuring their antioxidant activity by the DPPH method, determining a 2.67-fold reduction for 3-*O*-α-d-glucosyl resveratrol and of 1.42-fold for 4′-*O*-α-d-glucosyl resveratrol [23]. The results described for the two 3-*O*-α-d-glucosyl resveratrol products agree with the one obtained for 3-*O*-ꞵ-d-xylobiosyl resveratrol in this work. However, other resveratrol glycoconjugates that bear the sugar motif in the same 3 or 4′ positions, such as piceid (3-*O*-β-D-glucosyl resveratrol, β- linkage instead of α-), 3-*O*-α-d-maltosyl resveratrol and 4′-*O*-α-d-maltosyl resveratrol (maltosyl motif instead of glucosyl), reduced the antioxidant activity to a different degree than their corresponding α-glucosides [22,23]. This suggests that the type of linkage (α or β) and the carbohydrate attached (glucose or maltose in this case) could also have an impact on the antioxidant capacity of the molecules, apart from the glycosylation position. Thus, more in-depth studies are needed to determine the role of the β-bond and xylobiose motif in the reduction in the antioxidant activity observed in the 3-*O*-ꞵ-d-xylobiosyl resveratrol. Furthermore, considering the controversy generated by other authors who affirm that it is the 4′ position and not the 3 that plays a more important role in the antioxidant activity of resveratrol [26,46], an exhaustive review of this type of assay should be carried out.

### 3.5. Study of the Solubility of 3-O-β-d-Xylobiosyl Resveratrol with Respect to Its Aglycone

The extremely low aqueous solubility of resveratrol is one of the most challenging bottlenecks in obtaining the maximum benefits of this bioactive compound [2,3]. Following our experimental setup, the solubility of resveratrol was found to be 0.038 ± 0.003 g·L^−1^, which is consistent with previous measurements [3,5,6]. In turn, for the 3-*O*-ꞵ-d-xylobiosyl resveratrol, the solubility was 184.627 g·L^−1^, 4866 times greater than that of the aglycone.

This novel xylobioside is the most soluble among the resveratrol glycosides known to date, even higher than the reported 3,5-*O*-ꞵ-d-diglucosyl resveratrol, which increased by 4300 times that of the non-glycosylated substrate [26]. The improvements in the solubility described for the other resveratrol glycosides are considerably lower. For example, for the 3-*O*-α-d-glucosyl resveratrol, 4′-*O*-α-d-glucosyl resveratrol and 3,4′-*O*-α-d-diglucosyl resveratrol, the solubility increased by 65 times and, for piceid (3-*O*-β-D-glucosyl resveratrol), only by 12.3 times compared to the aglycon [22]. These data also show that the anomeric configuration (α or ꞵ) of the glycosidic linkage influences the solubility of the glycosides in aqueous media while presenting glycosylation in the 3 or 4′ position does not affect it that much. This hypothesis is supported by other authors who reported solubilities of 3.8-fold, 14.3-fold and 13.1-fold higher than that of the aglycone for piceid, 4′-*O*-α-d-glucosyl and 3-*O*-α-d-glucosyl resveratrol, respectively [24]. In another work, piceid exhibited a 31.5-fold improvement in solubility and the natural resveratrolside (4′-*O*-ꞵ-d-glucosyl resveratrol) a 35.5-fold increase [26]. In the case of our 3-O-ꞵ-D-xylobiosyl resveratrol, it seems that a longer sugar array, such as xylobiose, is also enhancing the aqueous solubility of resveratrol.

### 3.6. Deglycosylation of 3-O-β-d-Xylobiosyl Resveratrol by Colonic Microbiota

Bacterial species living in the small intestine include mainly lactobacilli and bifidobacteria that are not active hydrolyzing xylose-based glycosides [35] and, therefore, they most likely cannot deglycosylate 3-*O*-ꞵ-d-xylobiosyl resveratrol. Consequently, this xylobioside will reach the large intestine, where it is expected to interact with residing bacteria, particularly with *Bacteroides* species, which are well-known degraders of xylose-containing fiber [35,55,56]. Then, free resveratrol will be released and, as previously reported, metabolized by human colonic microbiota, leading to dihydroresveratrol, 3,4′-dihydroxy-*trans*-stilbene and 3,4′-dihydroxybibenzyl (lunularin), which are also bioactive metabolites with potential health benefits that can be absorbed in the large intestine [35,36,37,38,39,40]. Thus, a preliminary study aiming to confirm whether colonic bacteria are capable of hydrolyzing 3-*O*-ꞵ-d-xylobiosyl resveratrol was carried out.

Changes in the precursor compound, 3-*O*-ꞵ-d-xylobiosyl resveratrol, and in the main microbial metabolite to which it should be hydrolyzed, *trans*-resveratrol, were monitored during in vitro colonic fermentation by using fecal samples from two healthy donors (Figure 5). Control experiments indicated that 3-*O*-ꞵ-d-xylobiosyl resveratrol was stable in the medium without bacteria during the first 6 h of incubation, although a slight decrease was observed after 24 h (data not shown). Incubation with colonic microbiota contained in the feces showed that 3-*O*-ꞵ-d-xylobiosyl resveratrol was completely hydrolyzed, producing *trans*-resveratrol after 4 h fermentation (volunteer 1) or in 6 h (volunteer 2). The concentration of resveratrol then decreased steadily, suggesting that the gut microbiota metabolized it into other compounds that were not identified in the present study. This downstream metabolic conversion was proved, as neither 3-*O*-ꞵ-d-xylobiosyl resveratrol nor *trans*-resveratrol were detected after 12 h of incubation with the feces from volunteer 2, who showed a faster metabolism, while a concentration of 10 μM *trans*-resveratrol was still observed in the samples from volunteer 1, although no resveratrol was detected after 24 h incubation (Figure 5). Interindividual differences in the metabolism of phenolic compounds in gut microbiota have already been described for other phenols including resveratrol [37,57].

These results indicate that bacterial colonic strains can deglycosylate the xylobioside, releasing free resveratrol. The fact that 3-*O*-ꞵ-d-xylobiosyl resveratrol could bypass the extensive metabolism that takes place in the small intestine and reach the colon, where it is deglycosylated, could enable a prolonged interaction over time of resveratrol with colonic microbiota and its conversion to other bioactive metabolites. Nevertheless, although this investigation lays the groundwork for future research, more in-depth studies should be performed to understand how this xylobioside acts in the large intestine.

## 4. Conclusions

This work has corroborated that the rational design of glycosyl hydrolases is a promising strategy to eliminate or minimize the hydrolytic activity of native enzymes, enabling an efficient synthesis of glycosides of phenolic compounds. The rXynSOS-E236G glycosynthase, a mutagenic variant developed from a GH10 endoxylanase of the ascomycete *T. amestolkiae*, was successfully used to glycosylate resveratrol, yielding 3-*O*-ꞵ-d-xylobiosyl resveratrol as the main product. To the best of our knowledge, there are no previous references to resveratrol glycosides with xylobiose, whether extracted from natural sources or chemically or enzymatically synthesized. The very high solubility of 3-O-ꞵ-d-xylobiosyl resveratrol could allow the ingestion of more concentrated doses of the active ingredient that would reach the distal part of the gastrointestinal tract. There, according to the preliminary results presented in this work, it would be deglycosylated by the colonic microbiota. Both features of this novel compound open a new horizon to improve the bioavailability and therapeutic applications of resveratrol.

## Figures and Tables

**Figure 1 antioxidants-12-00085-f001:**
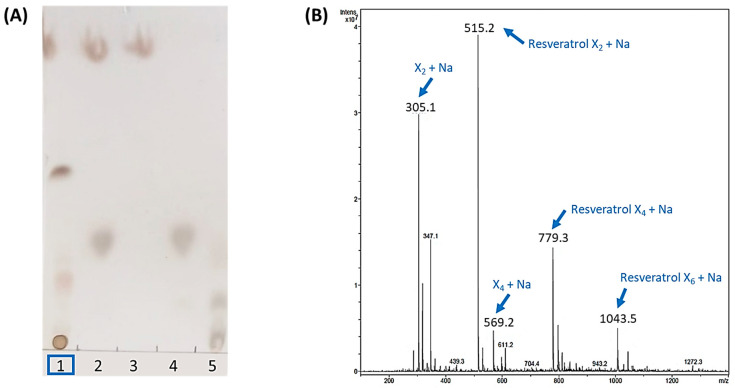
Analysis of the standard resveratrol glycosylation reaction catalyzed by rXynSOS-E236G glycosynthase using X_2_F as the donor. (**A**) Thin layer chromatography (TLC). Lane 1: sample of the glycosylation mixture, containing resveratrol, X_2_F, the catalyst and the reaction products. Lane 2: negative control with resveratrol and X_2_F and no catalyst. Lane 3: negative control containing only the resveratrol acceptor. Lane 4: negative control consisting of only the X_2_F donor. Lane 5: standards’ mixture with xylose, xylobiose, xylotriose and xylotetraose. (**B**) ESI-MS spectrum (positive mode). The *m*/*z* of ions corresponding to the Na^+^ adducts of the resveratrol glycosides, xylobiose and xylotetraose are indicated with blue arrows. Resveratrol was not detected in the positive mode spectrum but was clearly visible in the negative mode spectrum (Appendix A).

**Figure 2 antioxidants-12-00085-f002:**
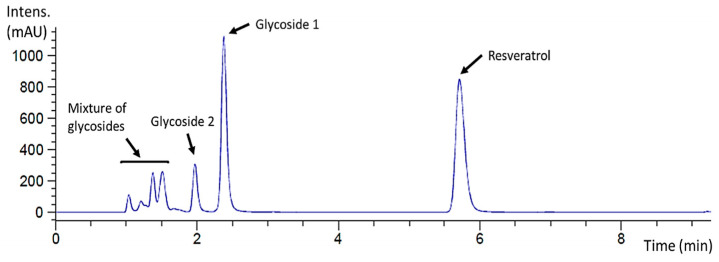
HPLC chromatogram (λ = 270 nm) of the standard resveratrol glycosylation reaction catalyzed by rXynSOS-E236G glycosynthase using X_2_F as the donor. The resveratrol used as the acceptor eluted at 5.71 min, the major peak (2.38 min) corresponding to a reaction product was labeled as glycoside **1** and the minor one (1.97 min) as glycoside **2**. A mixture of potential resveratrol glycosides eluted earlier.

**Figure 3 antioxidants-12-00085-f003:**
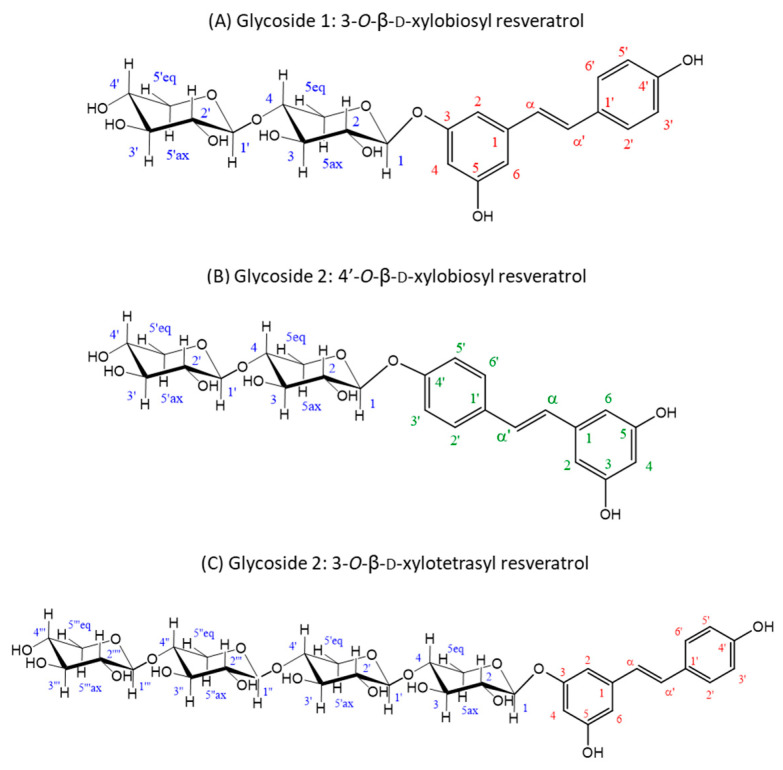
Structures deduced from NMR analysis of the purified glycosides **1** and **2**. (**A**) Glycoside **1** is the major product of the glycosylation reaction and was identified as 3-*O*-ꞵ-d-xylobiosyl resveratrol. (**B**,**C**) Glycoside **2** contains a mixture of minor reaction products resulting from the glycosylation of resveratrol, identified as 4′-*O*-ꞵ-d-xylobiosyl resveratrol (**B**) and 3-*O*-ꞵ-d-xylotetraosyl resveratrol (**C**) in a 3:1 ratio. Every C atom and associated proton in the molecules are numbered to clarify the identification of the signals (xylobiose C atoms are represented in blue, resveratrol C atoms of 3-*O*-ꞵ-d-xylobiosyl resveratrol and 3-*O*-ꞵ-d-xylotetraosyl resveratrol in red and resveratrol C atoms of 4′-*O*-ꞵ-d-xylobiosyl resveratrol in green) in the NMR spectra (Appendix A).

**Figure 4 antioxidants-12-00085-f004:**
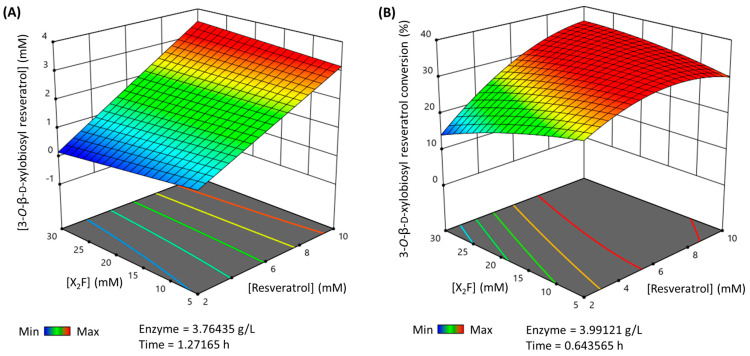
3D representation of the optimized reactions for the synthesis of 3-*O*-ꞵ-d-xylobiosyl resveratrol catalyzed by rXynSOS-E236G glycosynthase. The reaction conditions were predicted by a multiparametric model obtained following a response surface method. (**A**) Maximum production of 3-*O*-ꞵ-d-xylobiosyl resveratrol. (**B**) Maximum conversion (%) of resveratrol to 3-*O*-ꞵ-d-xylobiosyl resveratrol. The color code represents the production/conversion (%) range from minimum (blue) to maximum (red).

**Figure 5 antioxidants-12-00085-f005:**
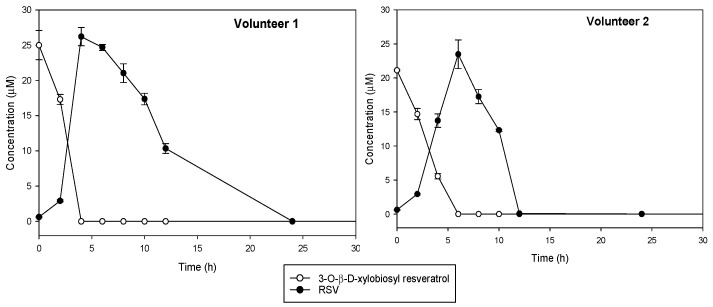
Kinetics of in vitro conversion under anaerobic conditions of 3-*O*-ꞵ-d-xylobiosyl resveratrol (white dots) into *trans*-resveratrol (RSV, black dots) by fecal microbiota from two different healthy volunteers.

**Table 1 antioxidants-12-00085-t001:** Antioxidant activity of resveratrol and its major xylobioside using ABTS^•+^ as substrate and Trolox as reference antioxidant compound.

Compound	R^2^	TEAC
Trolox	0.993	1
Resveratrol	0.997	0.472 ± 0.039
3-*O*-ꞵ-**d**-xylobiosyl resveratrol	0.999	1.042 ± 0.05 *

Data expressed as mean ± SD (*n* = 3, * *p* < 0.001 vs. resveratrol).

## Data Availability

Data are contained within the article and Appendix A.

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
