# Peer review of "Synthesis and Characterization of a Novel Resveratrol Xylobioside Obtained Using a Mutagenic Variant of a GH10 Endoxylanase"

_antioxidants, 2022, doi:10.3390/antiox12010085_

Round 1

Reviewer 1 Report

This is a very interesting manuscript; however, the author made a very serious mistake in the experimental design, as follows, the data from the Colonic fermentation only support the metabolism but not absorption. The authors must be performed in vitro absorption assay and in vitro intestinal permeation assays to demonstrate resveratrol absorption.

Author Response

We understand the concerns of the reviewer about the biological activity tests performed with resveratrol xylobioside and agree that more experiments are needed to demonstrate resveratrol absorption in the colon. However, we would like to emphasize that this experiment is outside the scope of our work. Our main objective was to demonstrate that resveratrol xylobiosides can be enzymatically produced by the mutagenic variant of a fungal endoxylanase, which has been confirmed. Furthermore, the major glycosylation product has been identified by NMR, its synthesis has been optimized and its properties have been studied. The xylobioside showed to be much more soluble than the aglycone and this would allow its administration in higher doses than free resveratrol. However, it must be hydrolyzed, releasing resveratrol to exert its biological action. The resveratrol glycoside is expected to reach the colon, since the microbiota in the small intestine cannot break it down, but colonic bacteria will probably do it. Hence, we designed a preliminary experiment to test whether the xylobioside could be hydrolyzed in the distal part of the gastrointestinal tract. This would allow the subsequent metabolism of resveratrol, which is thoroughly described in the literature. We have added several references to support this statement (line 100; lines 513-514). In addition, according to your suggestion, we have changed the term "absorbed" to "metabolized" throughout the manuscript (lines 97-101; lines 552-555).

Reviewer 2 Report

The paper is very interesting but before its publication needs to suffer minor revision.

Authors introduce very well the problems about the scarce solubility of resveratrol in aqueous solution so resorting to prepare its xylobioside derivative can represent a good solution. This aim is reached by using a mutagenic variant of a GH10 endoxylanase, that represents the strength of the work, even if the cost of the process needs to be attended to.

Another important point evidenced by authors concerns the type of linkage (α or β) and the carbohydrate attached (glucose or maltose in this case) that could also have an impact in the  antioxidant capacity of the molecules.

Many techniques based on ESI-MS such as NMR methods were applied to characterize the obtained derivatives and make more value to the work.

Authors must act in general on the whole manuscript to better rationalize it, avoiding repeating the same concept very often. At this purpose, the introduction must be revised to also reduce its length to speed up the reading.

Author Response

We acknowledge the positive comments of the reviewer. The manuscript has been revised to rationalize it, removing unnecessary repetitions. Moreover, the introduction has been reduced as requested.